# TRAJECTORY-CONSISTENT FLOWS: ENABLING FAST SAMPLING FOR FLOW MATCHING MODELS

## ABSTRACT

Diffusion and flow matching models have recently achieved remarkable generative performance, but their reliance on iterative ODE or SDE solvers results in slow and computationally expensive sampling. In this work, we introduce *Trajectory-Consistent Flows* (TCF), a framework that unifies efficient training and accelerated sampling through a Taylor-expansion-based formulation. TCF jointly optimizes a flow matching model $p_\theta$ and a fast-sampling surrogate $q_\theta$ via a unified objective. We construct $q_\theta$ using a second-order Taylor expansion as a trajectory-consistent approximation of $p_\theta$'s ODE flow, enabling high-fidelity generation with as few as 5 sampling steps. We further extend this idea to a third-order expansion, achieving additional performance gains without increasing computational cost. With further architectural and training enhancements, TCF achieves significantly improved sampling quality while retaining fast and stable training, making it particularly suitable for real-time generative applications.

## 1 INTRODUCTION

Generative modeling aims to approximate the true data distribution $p_{\text{data}}(\boldsymbol{x})$ by learning a parameterized model distribution $p_\theta(\boldsymbol{x})$. Since $p_{\text{data}}(\boldsymbol{x})$ is typically unknown, direct minimization of the Kullback–Leibler (KL) divergence between $p_\theta(\boldsymbol{x})$ and $p_{\text{data}}(\boldsymbol{x})$ is infeasible. Instead, *Maximum Likelihood Estimation* (MLE) provides a tractable surrogate by maximizing the log-likelihood over observed samples drawn from an empirical distribution $\hat{p}_{\text{data}}(\boldsymbol{x})$ (Espuña I Fontcuberta, 2022). This procedure corresponds to minimizing the KL divergence $D_{\text{KL}}(\hat{p}_{\text{data}}(\boldsymbol{x}) \parallel p_\theta(\boldsymbol{x}))$, thereby encouraging $p_\theta(\boldsymbol{x})$ to align closely with $\hat{p}_{\text{data}}(\boldsymbol{x})$:

$$\min_\theta D_{\text{KL}}(\hat{p}_{\text{data}}(\boldsymbol{x}) \parallel p_\theta(\boldsymbol{x})) = \min_\theta \left[ \mathbb{E}_{\boldsymbol{x} \sim \hat{p}_{\text{data}}} \left[ \log \hat{p}_{\text{data}}(\boldsymbol{x}) \right] - \mathbb{E}_{\boldsymbol{x} \sim \hat{p}_{\text{data}}} \left[ \log p_\theta(\boldsymbol{x}) \right] \right] \quad (1)$$

$$\propto \max_\theta \mathbb{E}_{\boldsymbol{x} \sim \hat{p}_{\text{data}}} \left[ \log p_\theta(\boldsymbol{x}) \right] \quad (2)$$

In recent years, expressive models such as diffusion models (Ho et al., 2020; Dhariwal & Nichol, 2021) and flow matching models (Lipman et al., 2023; Liu et al., 2023) have emerged. These models achieve state-of-the-art results across images, audio, and text, producing highly realistic samples. Although they do not directly optimize the log-likelihood, their objectives can be interpreted as variational bounds or approximations to the data likelihood, and empirical evidence suggests they capture $\hat{p}_{\text{data}}(\boldsymbol{x})$ effectively. From a theoretical perspective, diffusion and flow matching can be seen as two sides of the same coin (Gao et al., 2024): diffusion models approximate the reverse-time stochastic process of a noisy forward dynamics, whereas flow matching models learn to match the conditional velocity for continuous transport. The primary drawback of both approaches is the slow, iterative sampling process, which requires many function evaluations for a single output. This inefficiency poses a significant barrier to real-time or resource-constrained applications, motivating ongoing research on acceleration techniques and alternative model formulations.(Song et al., 2023; Esser et al., 2024; Lu et al., 2022; Kim et al., 2023).

In this work, rather than pursuing a more accurate estimation of the data likelihood that simultaneously ensures both efficient sampling and high-fidelity generation, we propose an alternative formulation. Specifically, we introduce a new model distribution $q_\theta$ as a fast estimator of $p_\theta$ and

propose to train our model using a joint training objective:

$$\min_{\theta} \ \mathcal{L}(\theta) = D_{\mathrm{KL}}(\hat{p}_{\mathrm{data}}(\boldsymbol{x}) \,\|\, p_{\theta}(\boldsymbol{x})) + \lambda \, D_{\mathrm{KL}}(p_{\theta}(\boldsymbol{x}) \,\|\, q_{\theta}(\boldsymbol{x})). \tag{3}$$

Here the first term corresponds to a flow matching objective that trains $p_{\theta}(\boldsymbol{x})$ to approximate $\hat{p}_{\mathrm{data}}(\boldsymbol{x})$ and the second term encourages $q_{\theta}(\boldsymbol{x})$ to align closely to $p_{\theta}(\boldsymbol{x})$. Both $p_{\theta}$ and $q_{\theta}$ share the same network parameters and are optimized jointly. In particular, we design $q_{\theta}$ to be independent of any ordinary differential equation (ODE) or stochastic differential equation (SDE) solver at sampling time, enabling fast inference. As a result, the overall framework combines the high-fidelity generation of flow matching with the sampling efficiency of $q_{\theta}$.

However, learning two model distributions together presents notable challenges. Since $p_{\theta}$ and $q_{\theta}$ typically rely on different inference mechanisms, they often require different model formulations, resulting in additional forward and backward passes. This significantly increases computational cost and memory usage. In addition, while the first KL term can be efficiently optimized using a flow matching loss, the second KL term is more difficult to handle in practice due to the lack of closed-form densities of $p_{\theta}$ and $q_{\theta}$. One feasible way to approximate the minimization of $D_{\mathrm{KL}}(p_{\theta}(\boldsymbol{x}) \,\|\, q_{\theta}(\boldsymbol{x}))$ is to encourage the outputs of the two model distribution to match under shared inputs. This aligns with the idea of knowledge distillation techniques (Hinton et al., 2015). However, sampling from $p_{\theta}$ during joint training still requires solving an ODE, which is computationally expensive and impractical. To overcome these difficulties, we propose an efficient framework that enables joint training of $p_{\theta}$ and $q_{\theta}$ without ODE simulations. In summary, our contribution is as follows:

- We propose a novel training framework that jointly optimizes a standard flow matching model $p_{\theta}$ and its fast-sampling surrogate $q_{\theta}$ through a unified objective, enabling efficient training and generation without iterative ODE or SDE solvers.

- We construct $q_{\theta}$ using a second-order Taylor expansion as a trajectory-consistent approximation of $p_{\theta}$'s ODE flow, and empirically show that it can match $p_{\theta}$'s performance with as few as 5 sampling steps.

- We extend this approach to a third-order Taylor expansion, which improves upon the second-order approximation effectively without increasing computational cost.

- We explore architectural design improvements, and demonstrate that, across multiple datasets, our model approaches the performance of state-of-the-art diffusion and flow matching models.

## 2 Flow matching and Rectified Flow

*Flow matching* (Lipman et al., 2023) is a recent generative modeling paradigm that formulates sample generation as learning a continuous-time flow transporting a simple base distribution (e.g., Gaussian noise) to $\hat{p}_{\mathrm{data}}(\boldsymbol{x})$. Unlike score-based diffusion models that rely on denoising score matching, flow matching methods directly learn the vector field of an ODE (or its velocity function) that defines the transport trajectory. The learning objective is constructed such that the model matches the flow of an optimal transport or probability path between the source and target distributions, typically using a loss function of the form:

$$\mathcal{L}_{\mathrm{FM}} = \mathbb{E}_{\boldsymbol{x} \sim \hat{p}_{\mathrm{data}}, t \sim \mathcal{U}(0,1)} \left[ \|v(\boldsymbol{x}, t) - \boldsymbol{v}_{\theta}(\boldsymbol{x}, t)\|^2 \right],$$

where $\mathcal{U}(\cdot, \cdot)$ represents uniform distribution, $v(\cdot, \cdot)$ and $\boldsymbol{v}_{\theta}(\cdot, \cdot)$ are the reference and learned velocity fields, respectively. This formulation enjoys stable training, (in some cases) exact likelihood evaluation, and interpretable dynamics (Lipman et al., 2023).

*Rectified Flow* (Liu et al., 2023), also known as *Stochastic Interpolants* (Albergo et al., 2023), is a specific instance of flow matching that improves the generation quality and the training efficiency by defining a "rectified" trajectory between noise and data samples. Instead of learning to match arbitrary transport paths, Rectified Flow defines a linear interpolation between a data sample $\boldsymbol{x}_1$ and its noisy counterpart $\boldsymbol{x}_0$, and constructs a target velocity field aligned with this interpolation. The model is trained to match this field across various intermediate time steps $t \in [0, 1]$, while generation is performed by solving the learned ODE in reverse. Notably, Rectified Flow achieves high sampling quality with fewer sampling steps and has become a competitive alternative to diffusion-based

Figure 1: Illustration of TCF, which jointly learns two distributions, $p_\theta$ and $q_\theta$. The training objective combines flow matching (to train $p_\theta$) with distribution alignment between $p_\theta$ and $q_\theta$. This alignment is achieved via three core constraints, each represented by colored arrows in the figure.

models. However, it still relies on numerical integration in learned flow with solvers such as RK45 (Virtanen et al., 2020), which incurs a high computational cost in inference time.

## 3    TRAJECTORY-CONSISTENT FLOWS

We begin by modeling the generative process $p_\theta$ as a flow matching model governed by a neural ODE parameterized by $\theta$:

$$\frac{d\boldsymbol{x}_t}{dt} = \boldsymbol{v}_\theta^{(p)}(\boldsymbol{x}_t, t), \quad \text{with } \boldsymbol{x}_t = (1-t)\boldsymbol{x}_0 + t\boldsymbol{x}_1, \quad \text{for } t \in [0, 1] \tag{4}$$

where $\boldsymbol{x}_0 \sim \mathcal{N}(\boldsymbol{0}, \boldsymbol{I})$ and $\boldsymbol{x}_1 \sim \hat{p}_{\text{data}}(\boldsymbol{x})$. This ODE defines a continuous trajectory that interpolates between Gaussian noise and real data with a linear reference velocity field.

To enable fast inference, we introduce a surrogate distribution $q_\theta$ to approximate the ODE solution defined by $p_\theta$. To ensure that $q_\theta$ faithfully approximates the behavior of $p_\theta$, we require the two models to produce identical outputs under the same input. This is equivalent to demanding that the trajectories generated by $q_\theta$ and the ODE flow induced by $p_\theta$ remain perfectly aligned given the same input state—an objective we refer to as *trajectory consistency*. Specifically, for any data point $\boldsymbol{x}_\tau$ at time $\tau \in [0, 1]$, with $t \in [\tau, 1]$ and $\boldsymbol{x}_t$ be an arbitrary point that lies along the ODE trajectory staring from $\boldsymbol{x}_\tau$, we define $f_\theta$, the trajectory function of $q_\theta$, to estimate $\boldsymbol{x}_t$ given $\boldsymbol{x}_\tau$ via:

$$f_\theta(\boldsymbol{x}_\tau, t) = \boldsymbol{x}_t \quad \text{and} \quad \boldsymbol{x}_t = \boldsymbol{x}_\tau + \int_\tau^t \boldsymbol{v}_\theta^{(p)}(\boldsymbol{x}_\xi, \xi)\, d\xi, \tag{5}$$

where $\boldsymbol{x}_\xi$ denotes the latent state at intermediate time $\xi$ along the same trajectory defined by the ODE. In particular, one-step generation can be achieved by evaluating $f_\theta(\boldsymbol{x}_0, 1)$. However, computing this integral during training is intractable. To address this, we propose three integration-free constraints that enable the learning of $f_\theta$ without relying on ODE solvers:

**Boundary Condition.**    Ensures that $f_\theta$ reduces to the identity map at its starting point:

$$f_\theta(\boldsymbol{x}_\tau, \tau) = \boldsymbol{x}_\tau. \tag{6}$$

**Self-Consistency.**    Enforces that a long-range transformation can be decomposed into intermediate steps along the same trajectory:

$$f_\theta(\boldsymbol{x}_\tau, t_2) = f_\theta(f_\theta(\boldsymbol{x}_\tau, t_1), t_2), \quad \forall\, t_1 \in [\tau, 1] \text{ and } t_2 \in [t_1, 1]. \tag{7}$$

**Velocity Consistency.**    Ensures that the time derivative of $f_\theta$ at the starting time aligns with the ODE's velocity:

$$\left.\frac{\partial f_\theta(\boldsymbol{x}_\tau, t)}{\partial t}\right|_{t=\tau} = \boldsymbol{v}_\theta^{(p)}(\boldsymbol{x}_\tau, \tau). \tag{8}$$

**Theorem 1.** *If $f_\theta$ satisfies all three constraints, then its trajectory aligns with the ODE trajectory governed by $p_\theta$, ensuring that $q_\theta$ converges to $p_\theta$ in a trajectory-consistent manner.*

The formal proof of Theorem 1 is provided in Appendix A.1. Importantly, these constraints are integration-free and thus avoid expensive numerical simulations during training. To better illustrate how our model jointly learns the distributions $p_\theta$ and $q_\theta$ through the proposed constraints, we provide a schematic visualization in Figure 1. This theoretical result forms the foundation of our training framework. All subsequent design choices and algorithmic developments build upon this principle.

## 3.1 FORMULATION OF $f_\theta$ AND TRAINING OBJECTIVES

The naive implementation of the above constraints can require multiple network evaluations. In this section, we propose an efficient parameterization of $f_\theta$ that supports fast and scalable training. We first parameterize the trajectory function $f_\theta : (\boldsymbol{x}_\tau, t) \mapsto \boldsymbol{x}_t$ using a second-order Taylor-like expansion:

$$f_\theta(\boldsymbol{x}_\tau, t) = \boldsymbol{x}_\tau + (t - \tau)\boldsymbol{v}_\theta(\boldsymbol{x}_\tau, t) + \frac{1}{2}(t - \tau)^2 \boldsymbol{u}_\theta(\boldsymbol{x}_\tau, t), \quad t \in [\tau, 1]. \tag{9}$$

where $\boldsymbol{v}_\theta$ and $\boldsymbol{u}_\theta$ are the two outputs of a shared neural network with separate heads. This formulation provides an explicit parametric structure for $f_\theta$, allowing its first- and second-order derivatives with respect to $t$ at $t = \tau$ to be computed analytically, without the need for numerical differentiation. Notice that, with this formulation, the *boundary condition* is also satisfied by design. We now describe our training objectives, which involve training both $p_\theta$ and $q_\theta$.

**Flow Matching Training.** As introduced earlier, $p_\theta$ is trained via the standard flow matching objective:

$$\mathcal{L}_{\mathrm{FM}} = \left\| \boldsymbol{v}_\theta^{(p)}(\boldsymbol{x}_\tau, \tau) - (\boldsymbol{x}_1 - \boldsymbol{x}_0) \right\|_2^2. \tag{10}$$

According to the *velocity consistency* constraint, we have $\boldsymbol{v}_\theta^{(p)}(\boldsymbol{x}_\tau, \tau) = \boldsymbol{v}_\theta(\boldsymbol{x}_\tau, \tau)$. In practice, we relax this to:

$$\boldsymbol{v}_\theta^{(p)}(\boldsymbol{x}_\tau, \tau) \approx \boldsymbol{v}_\theta(\boldsymbol{x}_\tau, t), \quad \text{for any } t \geq \tau, \tag{11}$$

then the flow matching objective could be rewritten as:

$$\mathcal{L}_{\mathrm{FM}} = \left\| \boldsymbol{v}_\theta(\boldsymbol{x}_\tau, t) - (\boldsymbol{x}_1 - \boldsymbol{x}_0) \right\|_2^2. \tag{12}$$

**Proposition 1.** *Suppose $p_\theta$ is trained using Eq. 12, then $\boldsymbol{v}_\theta$ satisfies:*

$$\boldsymbol{v}_\theta(\boldsymbol{x}_\tau, t) \equiv \boldsymbol{v}_\theta(\boldsymbol{x}_\tau, \tau), \quad \forall\, t \in [\tau, 1]. \tag{13}$$

*Thus, the **velocity consistency** constraint is satisfied.*

This invariance emerges because the optimal prediction of $\boldsymbol{v}_\theta$ is primarily determined by the input $\boldsymbol{x}_\tau$ and the network weights, while the time input $t$ should not affect the learned dynamics. In other words, $\boldsymbol{v}_\theta$ represents the instantaneous velocity at the starting position $\boldsymbol{x}_\tau$. A detailed proof of Proposition 1 is in Appendix A.2.

**Self-Consistency Training.** Given that the *boundary condition* and *velocity consistency* constraints are satisfied by construction, the training of $q_\theta$ focuses on enforcing *self-consistency*. Specifically, we minimize the discrepancy between the direct and two-step compositions reaching the same target time $t_2$:

$$\mathcal{L}_{\mathrm{consist}} = \left\| \frac{f_\theta(\boldsymbol{x}_\tau, t_2) - \mathrm{sg}[f_\theta(f_\theta(\boldsymbol{x}_\tau, t_1), t_2)]}{t_2 - \tau} \right\|_2^2. \tag{14}$$

Here $\mathrm{sg}[\cdot]$ denotes the stop-gradient operator that blocks gradient flow through the reference trajectory. The $(t_2 - \tau)$ normalization ensures scale-invariant learning across different time intervals. This loss stimulates the function values at $t_2$ to remain consistent across various trajectory decompositions.

To optimize $f_\theta$ under the proposed constraints, we design the training process outlined in Algorithm 1, referred to as *Alg-A*. This procedure incorporates all loss terms and requires three forward passes—only one of which involves gradient computation—followed by a single backward pass, enabling effective and reasonably efficient training. While *Alg-A* yields strong empirical results, we also develop a lightweight alternative that further reduces training computational cost. Specifically, enforcing the *Self-consistency* constraint in its original form typically involves three evaluations of $f_\theta$. However, we observe that this constraint can be relaxed to the following local condition:

**Corollary 1.** *Let $\Delta t$ be a small time increment and $f_\theta(\boldsymbol{x}_\tau, t)$ be a trajectory function that satisfies the following local consistency condition:*

$$f_\theta(\boldsymbol{x}_\tau, t) = f_\theta(\boldsymbol{x}_{\tau+\Delta t}, t), \quad \text{where } \boldsymbol{x}_{\tau+\Delta t} = f_\theta(\boldsymbol{x}_\tau, \tau + \Delta t). \tag{15}$$

*Then, $f_\theta$ also satisfies the global* Self-Consistency *constraint for any $t \in [\tau + \Delta t, 1]$:*

$$f_\theta(\boldsymbol{x}_\tau, t_2) = f_\theta(f_\theta(\boldsymbol{x}_\tau, t_1), t_2) \quad \forall\, t_1 \in [\tau, t_2]. \tag{16}$$

Intuitively, this constraint states that if two points are infinitesimally close along the ODE path, their mappings to a future time $t$ should agree. We prove in Appendix A.3 that training with this local constraint is equivalent to enforcing the original *Self-consistency* constraint. However, direct optimization of Eq. 15 still requires three forward passes. To reduce this cost, we approximate $\boldsymbol{x}_{\tau+\Delta t}$ using the first-order Taylor expansion of $f_\theta$ at time $\tau$:

$$f_\theta(\boldsymbol{x}_\tau, \tau + \Delta t) = \boldsymbol{x}_\tau + \Delta t \boldsymbol{v}_\theta(\boldsymbol{x}_\tau, \tau + \Delta t) + \mathcal{O}(\Delta t^2) \approx \boldsymbol{x}_\tau + \Delta t \boldsymbol{v}_\theta(\boldsymbol{x}_\tau, t). \tag{17}$$

Here, we reuse $t$ in place of $\tau + \Delta t$ as the target time for efficiency, under the assumption of Eq. 13. This approximation reduces the training overhead by eliminating one forward evaluation and leads to a more efficient training algorithm, denoted as *Alg-B* (see Algorithm 2).

---

**Algorithm 1** Training algorithm A

**Input:** dataset $\mathcal{D}$, initial model $\mathcal{F}_\theta$, learning rate $\eta$
**repeat**
  $\boldsymbol{x}_1 \sim \mathcal{D}$
  $\boldsymbol{x}_0 \sim \mathcal{N}(\boldsymbol{0}, \boldsymbol{I})$
  $\tau, t_1, t_2 \leftarrow \text{sample\_times}()$   ▷ $\tau < t_1 < t_2$
  $\boldsymbol{x}_\tau \leftarrow \tau\boldsymbol{x}_1 + (1-\tau)\boldsymbol{x}_0$
  **with** no gradient
    $\boldsymbol{x}_{t_1} \leftarrow f_\theta(\boldsymbol{x}_\tau, t_1)$   ▷ Eq. 9
    $\boldsymbol{x}_{t_2} \leftarrow f_\theta(\boldsymbol{x}_{t_1}, t_2)$
  $\hat{\boldsymbol{x}}_{t_2}, \boldsymbol{v} \leftarrow f_\theta(\boldsymbol{x}_\tau, t_2)$
  $\mathcal{L}_{\text{FM}} \leftarrow \|\boldsymbol{v} - (\boldsymbol{x}_1 - \boldsymbol{x}_0)\|_2^2$
  $\mathcal{L}_{\text{consist}} \leftarrow \left\| \frac{\boldsymbol{x}_{t_2} - \hat{\boldsymbol{x}}_{t_2}}{t_2 - \tau} \right\|_2^2$
  $\mathcal{L}_{all} \leftarrow \lambda_1 \mathcal{L}_{\text{FM}} + \lambda_2 \mathcal{L}_{\text{consist}}$
  $\theta \leftarrow \eta \nabla \mathcal{L}_{all}$
**until** convergence

**Algorithm 2** Training algorithm B

**Input:** dataset $\mathcal{D}$, initial model $\mathcal{F}_\theta$, learning rate $\eta$, small time interval $\Delta t$
**repeat**
  $\boldsymbol{x}_1 \sim \mathcal{D}, \boldsymbol{x}_0 \sim \mathcal{N}(\boldsymbol{0}, \boldsymbol{I})$
  $\tau, t_2 \leftarrow \text{sample\_times}()$   ▷ $\tau + \Delta t < t_2$
  $t_1 \leftarrow \tau + \Delta t$
  $\boldsymbol{x}_\tau \leftarrow \tau\boldsymbol{x}_1 + (1-\tau)\boldsymbol{x}_0$
  $\hat{\boldsymbol{x}}_{t_2}, \boldsymbol{v} \leftarrow f_\theta(\boldsymbol{x}_\tau, t_2)$   ▷ Eq. 9
  **with** no gradient
    $\boldsymbol{x}_{t_1} \leftarrow \boldsymbol{x}_\tau + \Delta t * \boldsymbol{v}$
    $\boldsymbol{x}_{t_2} \leftarrow f_\theta(\boldsymbol{x}_{t_1}, t_2)$
  $\mathcal{L}_{\text{FM}} \leftarrow \|\boldsymbol{v} - (\boldsymbol{x}_1 - \boldsymbol{x}_0)\|_2^2$
  $\mathcal{L}_{\text{consist}} \leftarrow \left\| \frac{\boldsymbol{x}_{t_2} - \hat{\boldsymbol{x}}_{t_2}}{t_2 - \tau} \right\|_2^2$
  $\mathcal{L}_{all} \leftarrow \lambda_1 \mathcal{L}_{\text{FM}} + \lambda_2 \mathcal{L}_{\text{consist}}$
  $\theta \leftarrow \eta \nabla \mathcal{L}_{all}$
**until** convergence

---

### 3.2 THIRD-ORDER TAYLOR EXTENSION

The second-order Taylor parameterization in Eq. 9 provides a tractable formulation of $f_\theta$, but we observe that the approximation gap between $q_\theta$ and $p_\theta$ remains non-negligible under few-step inference. Among the three constraints introduced earlier, both the boundary condition and velocity consistency are satisfied by construction, and our experiments further confirm that $p_\theta$ itself performs well. This suggests that the remaining limitation primarily stems from the self-consistency loss in Eq. 14. While Eq. 14 enforces trajectory compositionality, it does not sufficiently constrain the local velocity field at later time steps, which may lead to mismatches during few-step sampling. We also experimented with alternative loss designs commonly used in generative modeling, such as perceptual losses (e.g., LPIPS) (Song et al., 2023; Liu et al., 2023) and Pseudo-Huber loss (Song & Dhariwal, 2024), but found that they did not offer significant improvements. This motivates us to seek a more principled way to incorporate self-consistency constraints.

Previously, our self-consistency constraint only ensured function-value consistency at $t_2$. A natural way to strengthen this constraint is to additionally require velocity consistency at $t_2$ from any start point $t_1$ on the same trajectory:

$$\left. \frac{\partial f_\theta(\boldsymbol{x}_\tau, t)}{\partial t} \right|_{t=t_2} = \left. \frac{\partial f_\theta(\boldsymbol{x}_{t_1}, t)}{\partial t} \right|_{t=t_2}. \tag{18}$$

While our model can principally compute the instantaneous velocity at $\boldsymbol{x}_{t_2}$, enforcing Eq. 18 requires additional function evaluations, which incur substantial computational overhead (detailed

explanation in Appendix B). Inspired by our earlier design where $\boldsymbol{v}_\theta$ and $\boldsymbol{u}_\theta$ share the same neural backbone, we extend this idea by introducing an additional output head such that computing $\boldsymbol{v}_\theta^*(\boldsymbol{x}_\tau, t_2)$, the instantaneous velocity at an arbitrary **target timestep** $t_2$ is straightforward. In particular, $\boldsymbol{v}_\theta^*(\cdot, \cdot)$ is explicitly designed to satisfy:

$$\boldsymbol{v}_\theta^*(\boldsymbol{x}_\tau, \tau) = \boldsymbol{v}_\theta(\boldsymbol{x}_\tau, \tau), \quad \frac{\partial^2 f_\theta(\boldsymbol{x}_\tau, t)}{\partial t^2}\bigg|_{t=\tau} = \lim_{t\to\tau} \frac{\boldsymbol{v}_\theta^*(\boldsymbol{x}_\tau, t) - \boldsymbol{v}_\theta(\boldsymbol{x}_\tau, t)}{t - \tau}. \tag{19}$$

Since the first- and second-order derivatives of $f_\theta$ at $\tau$ are already determined by the instantaneous velocity at the starting timestep ($\boldsymbol{v}_\theta$) and the ending timestep ($\boldsymbol{v}_\theta^*$), we extend the model to a third-order Taylor expansion to provide additional flexibility in trajectory modeling while retaining the first- and second-order constraints:

$$f_\theta(\boldsymbol{x}_\tau, t) = \boldsymbol{x}_\tau + (t - \tau)\boldsymbol{v}_\theta(\boldsymbol{x}_\tau, t) + \tfrac{1}{2}(t - \tau)^2 \boldsymbol{u}_\theta(\boldsymbol{x}_\tau, t) + \tfrac{1}{6}(t - \tau)^3 \boldsymbol{w}_\theta(\boldsymbol{x}_\tau, t), \tag{20}$$

where $\boldsymbol{w}_\theta$ denotes an additional neural head that captures third-order dynamics. Under this formulation, $\boldsymbol{v}_\theta^*(\boldsymbol{x}_\tau, t)$ can be parameterized as:

$$\boldsymbol{v}_\theta^*(\boldsymbol{x}_\tau, t) = \boldsymbol{v}_\theta(\boldsymbol{x}_\tau, t) + (t - \tau)\boldsymbol{u}_\theta(\boldsymbol{x}_\tau, t). \tag{21}$$

This design ensures that Eq. 19 is naturally satisfied.

**Proposition 2.** *Suppose the model $\theta$ is trained using the following loss:*

$$\mathcal{L}_{vec} = \left\| \boldsymbol{v}_\theta^*(\boldsymbol{x}_\tau, t_2) - \boldsymbol{v}_\theta^*(\boldsymbol{x}_{t_1}, t_2) \right\|_2^2. \tag{22}$$

*Then $\boldsymbol{v}_\theta^*$ corresponds to the instantaneous velocity at the target timestep, and such velocities are simultaneously guaranteed to remain self-consistent across decompositions.*

The formal proof of Proposition 2 is presented in Appendix A.4. In principle, the loss in Eq. 22 is sufficient to guarantee the original self-consistency constraint. However, in practice, we detach the gradient of $\boldsymbol{v}_\theta^*(\boldsymbol{x}_{t_1}, t_2)$ to significantly reduce both computation and memory overhead. While this approximation makes training feasible at scale, it prevents the third-order term $\boldsymbol{w}_\theta$ from receiving gradient signals. To compensate for this limitation, we retain both the original self-consistency loss (Eq. 14) and the new velocity loss in our final objective. The detailed training process is presented in Algorithm 3, referred to as *Alg-C*.

---

**Algorithm 3** Training algorithm C

**Input:** dataset $\mathcal{D}$, initial model $\mathcal{F}_\theta$, learning rate $\eta$
**repeat**
    $\boldsymbol{x}_1 \sim \mathcal{D}, \boldsymbol{x}_0 \sim \mathcal{N}(\boldsymbol{0}, \boldsymbol{I})$
    $\tau, t_1, t_2 \leftarrow \text{sample\_times}()$       $\triangleright \tau < t_1 < t_2$
    $\boldsymbol{x}_\tau \leftarrow \tau\boldsymbol{x}_1 + (1 - \tau)\boldsymbol{x}_0$
    **with** no gradient
        $\boldsymbol{x}_{t_1} \leftarrow f_\theta(\boldsymbol{x}_\tau, t_1)$          $\triangleright$ Eq. 20
        $\boldsymbol{x}_{t_2}, \boldsymbol{v}^{(1)}, \boldsymbol{u}^{(1)} \leftarrow f_\theta(\boldsymbol{x}_{t_1}, t_2)$
    $\hat{\boldsymbol{x}}_{t_2}, \boldsymbol{v}, \boldsymbol{u} \leftarrow f_\theta(\boldsymbol{x}_\tau, t_2)$
    $\mathcal{L}_{\text{FM}} \leftarrow \|\boldsymbol{v} - (\boldsymbol{x}_1 - \boldsymbol{x}_0)\|_2^2$
    $\mathcal{L}_{\text{consist}} \leftarrow \left\| \frac{\boldsymbol{x}_{t_2} - \hat{\boldsymbol{x}}_{t_2}}{t_2 - \tau} \right\|_2^2$
    $\mathcal{L}_{\text{vec}} \leftarrow \|\boldsymbol{v} + (t_2 - \tau)\boldsymbol{u} - (\boldsymbol{v}^{(1)} + (t_2 - t_1)\boldsymbol{u}^{(1)})\|_2^2$
    $\mathcal{L}_{all} \leftarrow \lambda_1 \mathcal{L}_{\text{FM}} + \lambda_2 \mathcal{L}_{\text{consist}} + \lambda_3 \mathcal{L}_{\text{vec}}$
    $\theta \leftarrow \eta \nabla \mathcal{L}_{all}$
**until** convergence

---

## 4 EXPERIMENTS

TCF naturally supports a variety of training strategies, and we provide detailed experimental results in the Appendix D.3. While strategies such as distillation or joint training with a pretrained model may yield stronger performance, we emphasize that end-to-end training remains more scalable and versatile. It not only facilitates straightforward extension to other datasets but also offers greater convenience for downstream adaptation like supervised fine-tuning (SFT). For these reasons, we prefer the end-to-end training paradigm as the default choice in our framework. Due to page limitations, we provide experimental details and extended results in Appendix D.

### 4.1 COMPARISON WITH BASELINE MODELS

We first evaluate our method on the CIFAR-10 dataset (Krizhevsky et al., 2009), adopting the default architecture and training settings from Rectified Flow (Liu et al., 2023) to ensure a controlled setup for direct comparison. In addition to our baseline, we also train two reference models: (i) a model optimized with the consistency training objective from (Song et al., 2023), denoted as *consistency*

Table 1: Comparison of the proposed model and baseline models.

| Model | Sampling Method | NFE ($\downarrow$) | FID ($\downarrow$) | IS ($\uparrow$) |
|---|---|---|---|---|
| RF (baseline) Liu et al. (2023) | Euler | 10 | 12.8 | 8.50 |
| | Euler | 100 | 3.30 | 9.38 |
| | Euler | 1000 | 2.74 | 9.52 |
| | RK45 | 116 | 2.62 | 9.50 |
| Consistency Training | - | 2 | 4.7 | 8.92 |
| Shortcut Model | - | 2 | 4.12 | 9.12 |
| TCF Alg-A (Alg-B) | Euler ($p_\theta$) | 10 | 12.8 (12.9) | 8.48 (8.53) |
| | Euler ($p_\theta$) | 100 | 3.34 (3.33) | 9.36 (9.55) |
| | Euler ($p_\theta$) | 1000 | 2.70 (2.78) | 9.56 (9.56) |
| | Second-Order ODE Solver | 10 | 3.58 (3.81) | 9.38 (9.12) |
| | Second-Order ODE Solver | 100 | 2.63 (2.81) | 9.71 (9.58) |
| | - | 1 | 6.72 (6.96) | 8.46 (8.32) |
| | - | 2 | 3.71 (3.68) | 9.31 ( 9.38) |
| | - | 5 | 2.69 (2.81) | 9.58 (9.58) |
| | - | 10 | 2.61 (2.64) | 9.68 (9.61) |
| TCF Alg-C | Second-Order ODE Solver | 10 | 3.60 | 9.38 |
| | Second-Order ODE Solver | 100 | 2.65 | 9.38 |
| | Third-Order ODE Solver | 5 | 2.98 | 9.31 |
| | Third-Order ODE Solver | 10 | 2.65 | 9.61 |
| | - | 1 | 3.98 | 9.24 |
| | - | 2 | 2.95 | 9.54 |
| | - | 5 | 2.61 | 9.58 |

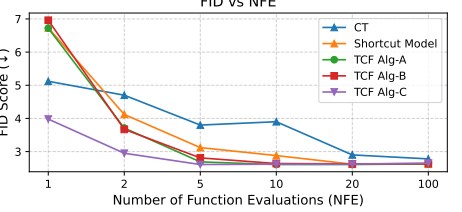 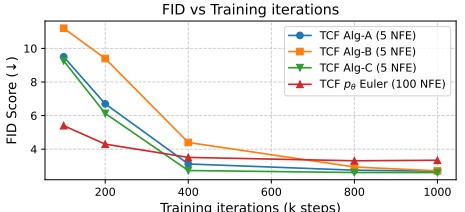

Figure 2: FID scores across various NFEs and training steps.

*training*, and (ii) a shortcut model (Frans et al., 2025) that follows the same trajectory design as ours. For both reference models, we strictly reuse our network architecture and training configuration, ensuring fairness in comparison. Details of the sampling algorithm are provided in Appendix C. Table 1 summarizes the quantitative results. When using first-order Euler sampling, our method achieves nearly identical performance to the baseline model in sampling from $p_\theta$, confirming that the joint training of $q_\theta$ does not interfere with the flow-matching dynamics of $p_\theta$. However, under a second-order ODE solver, our approach significantly outperforms the baseline with the same number of function evaluations (NFE). This highlights the strength of our framework: the Taylor-based formulation of $f_\theta$ grants access to higher-order derivatives, enabling more accurate integration beyond conventional flow matching setups. Moreover, few-step sampling via $q_\theta$ attains sample quality comparable to full Rectified Flow simulation while requiring far fewer steps, demonstrating that $q_\theta$ effectively approximates the trajectory induced by $p_\theta$. In particular, we find that a third-order Taylor expansion yields a substantial improvement over the second-order version, further enhancing sample quality without requiring additional NFEs. Finally, our fast sampler not only surpasses the shortcut baseline but also consistently outperforms the *consistency training* model (Song et al., 2023), showing that our method benefits from both stronger modeling fidelity and improved training efficiency. Further results across different training steps and NFEs are presented in Figure 2.

## 4.2 COMPUTATION COST AND TRAINING STABILITY

Our training procedure requires only one gradient-tracked forward pass and one backward pass per step—consistent with standard flow matching methods. The additional overhead arises from one (in

*Alg-B*) or two (in *Alg-A, C*) extra forward passes used for enforcing self-consistency, resulting in a modest 30%–50% increase in per-step runtime. Despite this, training remains highly efficient and stable. Across all experiments, we observe no signs of divergence or numerical instability.

## 4.3 EXPLORING DESIGN CHOICES

While the preceding configuration yields promising results, it still lags behind state-of-the-art (SOTA) generative models (Karras et al., 2022) in terms of sample quality, particularly as measured by FID. In this subsection, we analyze the root causes of this performance gap and propose targeted improvements. We identify two main factors contributing to the observed discrepancy:1) **architectural gap between $p_\theta$ and SOTA diffusion models:** The architecture inherited from Rectified Flow lacks the representational capacity and training sophistication of modern designs such as those used in diffusion models (Karras et al., 2022). And 2) **approximation gap between $q_\theta$ and $p_\theta$:** Even if $p_\theta$ produces high-quality samples, insufficient approximation by $q_\theta$ may degrade fast sampling performance. To systematically address both issues, we adopt a two-stage strategy: first enhancing the architecture and training of $p_\theta$, then improving the approximation quality of $q_\theta$.

**Enhancing the flow matching model $p_\theta$.** We improve $p_\theta$ through both architectural and training enhancements. Architecturally, we integrate key design components that strengthen model expressiveness. We adopt the log-normal time distribution from EDM, mapped to $[0, 1]$ via a sigmoid transformation, introduce input weighting regularized by $1/\sqrt{(1 - t)^2 + t\sigma_{\text{data}}^2}$, and employ larger batch sizes (from 512 to 2048) with extended training (from 400k to 800k steps). Together, these refinements reduce the FID on CIFAR-10 to 2.10, substantially narrowing the gap with diffusion models.

**Refining the fast sampler $q_\theta$.** With a stronger $p_\theta$ in place, we focus on reducing the mismatch between $q_\theta$ and $p_\theta$. We find that several factors influence the final performance of $p_\theta$, including the choice of time distributions ($t_1$ and $t_2$), the use of absolute versus relative time conditioning, the design of the $\ell_2$ loss, and the weighting of different loss terms. Each of these components plays a critical role in maintaining trajectory consistency and achieving high-fidelity few-step sampling. For clarity and reproducibility, we provide detailed implementation and hyperparameter specifications in Appendix D.2, where we systematically evaluate the effect of these factors on sampling quality and training stability.

Table 2: Reuslts on CIFAR-10 (unconditional).

| METHOD | NFE (↓) | FID (↓) |
|---|---|---|
| **Fast samplers & distillation models** | | |
| DPM-solver-fast (Lu et al., 2022) | 10 | 4.70 |
| 3-DEIS (Zhang & Chen, 2023) | 10 | 4.17 |
| UniPC (Zhao et al., 2023) | 10 | 3.87 |
| DFNO (LPIPS) (Zheng et al., 2023) | 1 | 3.78 |
| 2-Rectified Flow (+distill) (Liu et al., 2023) | 1 | 4.85 |
| TRACT (Berthelot et al., 2023) | 2 | 3.32 |
| Diff-Instruct (Luo et al., 2023) | 1 | 4.53 |
| CD (LPIPS) (Song et al., 2023) | 2 | 2.93 |
| SIM (Luo et al., 2024) | 1 | 2.06 |
| **Direct Generation** | | |
| Score SDE (Song et al., 2021b) | 2000 | 2.38 |
| DDPM (Ho et al., 2020) | 1000 | 3.17 |
| LSGM (Vahdat et al., 2021) | 147 | 2.10 |
| PFGM (Xu et al., 2022) | 110 | 2.35 |
| EDM (Karras et al., 2022) | 35 | 1.97 |
| NVAE (Vahdat & Kautz, 2020) | 1 | 23.5 |
| Glow (Kingma & Dhariwal, 2018) | 1 | 48.9 |
| MeanFlow (Geng et al., 2025a) | 1 | 2.92 |
| IMM (Zhou et al., 2025) | 2 | 1.98 |
| **Consistency model family** | | |
| CT (LPIPS) (Song et al., 2023) | 2 | 5.83 |
| iCT (Song & Dhariwal, 2024) | 2 | 2.46 |
| ECM (Geng et al., 2025b) | 2 | 2.11 |
| sCM (Lu & Song, 2025) | 2 | 2.06 |
| **Ours** | | |
| TCF (Alg-C) | 1 | 3.65 |
| TCF (Alg-C) | 2 | 2.45 |
| TCF (Alg-C) | 5 | 2.28 |
| TCF (Alg-C) | 10 | 2.10 |

As summarized in Table 2 and Table 3, the TCF model achieves competitive results with only 2 or 5 sampling steps, highlighting its efficiency in generation. Compared to recent methods in the Consistency Model family, TCF shows slightly lower performance in FID. However, these stronger baselines often come at the cost of increased training or inference complexity. For example, TCM (Lee et al., 2025) requires multi-stage training, and sCM (Lu & Song, 2025) depends on expensive Jacobian-vector product (JVP) computations. In contrast, TCF maintains a simpler, single-stage training process with good stability and lower memory overhead, making it more practical for real-world applications.

## 5 RELATED WORK

A general strategy for accelerating generative models is *self-distillation* explicitly or implicitly, where a multi-step teacher model and a fast student model are jointly learned in a single training process. *Consistency Models* (CMs) (Song et al., 2023) follow this paradigm by using a single network and a single loss to enforce temporal consistency, thereby enabling few-step or even single-step sampling. Despite their sampling efficiency, however, CMs face significant training challenges: they often require substantially longer training time compared to diffusion models of similar quality, and involve intricate hyperparameter tuning, which hinders their practical adoption. These limitations have motivated subsequent works (Geng et al., 2025b; Song et al., 2023; Lee et al., 2025; Lu & Song, 2025) that aim to improve stability, efficiency, and overall performance. *Consistency Trajectory Models* (CTMs) (Kim et al., 2023) extend the self-distillation idea by introducing separate networks and separate objectives for the teacher and student, while relying on repeated ODE solver calls during training. Although this improves flexibility, the heavy solver dependence increases training cost and introduces solver-induced errors, which degrade performance in the absence of adversarial

Table 3: Results on ImageNet $64 \times 64$ (class-conditional).

| METHOD | NFE (↓) | FID (↓) |
|---|---|---|
| **Fast samplers & distillation models** | | |
| DDIM (Song et al., 2021a) | 10 | 18.3 |
| DPM solver (Lu et al., 2022) | 10 | 7.93 |
| DEIS (Zhang & Chen, 2023) | 10 | 6.65 |
| DFNO (LPIPS) (Zheng et al., 2023) | 1 | 7.83 |
| TRACT (Berthelot et al., 2023) | 2 | 4.97 |
| Diff-Instruct (Luo et al., 2023) | 1 | 5.57 |
| CD (LPIPS) (Song et al., 2023) | 2 | 4.70 |
| **Direct Generation** | | |
| DDPM (Ho et al., 2020) | 250 | 11.0 |
| iDDPM (Nichol & Dhariwal, 2021) | 250 | 2.92 |
| ADM (Dhariwal & Nichol, 2021) | 250 | 2.07 |
| EDM (Karras et al., 2022) | 511 | 1.36 |
| EDM (Heun) (Karras et al., 2022) | 79 | 2.44 |
| **Consistency model family** | | |
| CT (LPIPS) (Song et al., 2023) | 2 | 11.1 |
| iCT (Song et al., 2023) | 2 | 3.20 |
| ECM-S (Geng et al., 2025b) | 2 | 2.79 |
| TCM (Lee et al., 2025) | 2 | 2.31 |
| sCM (Lu & Song, 2025) | 2 | 1.48 |
| **Ours** | | |
| TCF Alg-C | 1 | 3.94 |
| TCF Alg-C | 2 | 2.94 |
| TCF Alg-C | 5 | 2.24 |
| TCF Alg-C | 10 | 2.18 |

training. More recent approaches such as *Shortcut Models* (Frans et al., 2025) and *MeanFlow* (Geng et al., 2025a) train both the teacher $p_\theta$ and the student $q_\theta$ with shared parameters, but apply different objectives to different samples within the same batch, which complicates optimization and reduces efficiency. In particular, *MeanFlow* requires Jacobian–vector product (JVP) computations, which not only incur high memory overhead but also demand specialized hardware support—many devices offer limited or inefficient JVP implementations, further constraining scalability and stability. *SplitMeanFlow* Guo et al., 2025 mitigates this issue by removing the JVP requirement; however, its effectiveness for image generation or training from scratch has not been demonstrated. A different direction, *IMM* (Zhou et al., 2025), extends CMs with an MMD-based loss, aligning teacher and student distributions at the distributional level rather than matching ODE trajectories directly, which results in more stable training and strong empirical performance. Compared with these methods, our framework jointly trains $p_\theta$ and $q_\theta$ through a unified objective that is applied consistently to every sample, simplifying optimization while ensuring both stability and efficiency.

We also include a discussion on various distillation strategies in Appendix E.

## 6 CONCLUSION

In this work, we introduced *Trajectory-Consistent Flows* (TCF), a novel generative framework that unifies efficient training and accelerated sampling through a Taylor-expansion-based formulation. Our framework jointly optimizes a flow matching model $p_\theta$ and its fast-sampling surrogate $q_\theta$ via a unified training objective, eliminating the need for iterative ODE or SDE solvers. We designed $q_\theta$ as a trajectory-consistent approximation of $p_\theta$'s ODE flow using a second-order Taylor expansion, which enables accurate generation with as few as 5 sampling steps. We further extended this idea to a third-order expansion, achieving additional performance gains without extra computational cost. Beyond the Taylor-based formulation, we explored architectural and training improvements and demonstrated that TCF approaches the performance of state-of-the-art diffusion and flow matching models across multiple datasets. Overall, our study highlights that principled higher-order modeling provides a powerful alternative to existing generative paradigms, combining the efficiency of flow-based approaches with the sample quality of diffusion models.

## 7 ETHICS AND BROADER IMPACT STATEMENT

Generative models have widespread applications, ranging from creative content generation to data augmentation and scientific simulation. Our proposed framework advances this field by significantly reducing the computational cost of high-fidelity sample generation, making it more accessible for deployment in real-time or resource-constrained environments. This may democratize generative technologies in areas such as mobile AI, interactive media, or edge computing.

However, as with all generative models, potential risks arise along with utilization. Fast and efficient generation may exacerbate the misuse of synthetic content in misinformation, deepfakes, or privacy-violating applications. Additionally, acceleration may lead to wider deployment without sufficient oversight. We encourage researchers and practitioners to apply this method responsibly and in accordance with ethical guidelines, particularly in contexts involving sensitive or human-centered data.

## 8 REPRODUCIBILITY

In this work, we propose TCF, a novel training framework that jointly optimizes a standard flow matching model and its fast-sampling surrogate through a unified objective. A detailed explanation of the algorithm itself is provided in Section 3, with theoretical proofs provided in Appendix A. Sampling strategies are introduced in Appendix C. Main experimental results are in Section 4, with more details and extended results presented in Appendix D.

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

# A  THEORETICAL RESULTS

## A.1  PROOF OF THEOREM 1

**Theorem 1** *If the trajectory function $f_\theta$ satisfies the Boundary Condition, Self-Consistency, and Velocity Consistency, then the trajectory it defines aligns with the ODE trajectory governed by $p_\theta$. As a result, the surrogate sampler $q_\theta$ becomes trajectory-consistent with the flow induced by $p_\theta$.*

*Proof.* Let $\boldsymbol{x}_t = f_\theta(\boldsymbol{x}_\tau, t)$ denote the output of the surrogate sampler starting from the input state $\boldsymbol{x}_\tau$ at time $\tau$. To prove trajectory consistency, we need to show that $f_\theta$ satisfies the same ODE as $p_\theta$, i.e.,

$$\frac{d\boldsymbol{x}_t}{dt} = \boldsymbol{v}_\theta^{(p)}(\boldsymbol{x}_t, t), \quad \forall\, t \in [\tau, 1]. \tag{23}$$

We begin by computing the time derivative of $f_\theta$:

$$\frac{\partial f_\theta(\boldsymbol{x}_\tau, t)}{\partial t} = \lim_{\Delta t \to 0} \frac{f_\theta(\boldsymbol{x}_\tau, t + \Delta t) - f_\theta(\boldsymbol{x}_\tau, t)}{\Delta t} \tag{24}$$

$$= \lim_{\Delta t \to 0} \frac{f_\theta(\boldsymbol{x}_\tau, t + \Delta t) - \boldsymbol{x}_t}{\Delta t}. \tag{25}$$

Using the *Self-Consistency* condition, we can write:

$$f_\theta(\boldsymbol{x}_\tau, t + \Delta t) = f_\theta(f_\theta(\boldsymbol{x}_\tau, t), t + \Delta t) = f_\theta(\boldsymbol{x}_t, t + \Delta t). \tag{26}$$

Recall the *Boundary Condition*, we have:

$$f_\theta(\boldsymbol{x}_t, t) = \boldsymbol{x}_t \tag{27}$$

Substituting Eq. 26 and 27 back to Eq. 25, we get:

$$\frac{\partial f_\theta(\boldsymbol{x}_\tau, t)}{\partial t} = \lim_{\Delta t \to 0} \frac{f_\theta(\boldsymbol{x}_t, t + \Delta t) - f_\theta(\boldsymbol{x}_t, t)}{\Delta t} \tag{28}$$

$$= \frac{\partial f_\theta(\boldsymbol{x}_t, t)}{\partial t} \tag{29}$$

By the *Velocity Consistency* condition, we have:

$$\left.\frac{\partial f_\theta(\boldsymbol{x}_t, t)}{\partial t}\right|_{\boldsymbol{x}_t = f_\theta(\boldsymbol{x}_\tau, t)} = \boldsymbol{v}_\theta^{(p)}(\boldsymbol{x}_t, t). \tag{30}$$

Thus, we conclude that:

$$\frac{d\boldsymbol{x}_t}{dt} = \frac{\partial f_\theta(\boldsymbol{x}_\tau, t)}{\partial t} = \boldsymbol{v}_\theta^{(p)}(\boldsymbol{x}_t, t), \tag{31}$$

which proves that $f_\theta$ produces a trajectory consistent with the ODE solution defined by $p_\theta$.  □

## A.2  PROOF OF PROPOSITION 1

**Proposition 1** *Suppose the model $p_\theta$ is trained using the following flow matching loss:*

$$\mathcal{L}_{FM} = \mathbb{E}_{\boldsymbol{x}_0, \boldsymbol{x}_1, \tau}\left[\|\boldsymbol{v}_\theta(\boldsymbol{x}_\tau, t) - \boldsymbol{v}_{target}\|^2\right], \tag{32}$$

*where $\boldsymbol{x}_\tau = (1 - \tau)\boldsymbol{x}_0 + \tau \boldsymbol{x}_1$, and $\boldsymbol{v}_{target} = \boldsymbol{x}_1 - \boldsymbol{x}_0$ is a reference velocity field that depends only on $\boldsymbol{x}_0, \boldsymbol{x}_1$. Then, the learned velocity field $\boldsymbol{v}_\theta$ satisfies the following invariance:*

$$\boldsymbol{v}_\theta(\boldsymbol{x}_\tau, t) \equiv \boldsymbol{v}_\theta(\boldsymbol{x}_\tau, \tau), \quad \forall\, t \in [\tau, 1]. \tag{33}$$

*As a result, the **velocity consistency** constraint is automatically satisfied.*

*Proof.* During training, the objective in Eq. 12 aims to minimize the discrepancy between the predicted velocity $\boldsymbol{v}_\theta(\boldsymbol{x}_\tau, t)$ and the reference velocity $\boldsymbol{v}_{\text{target}}$, which is independent of $t$. That is, the supervision signal does not change with the value of $t$ and remains fixed for a given $\boldsymbol{x}_\tau$.

Therefore, under sufficient optimization, the model has no incentive to vary its output with respect to $t$; the optimal solution will be invariant in $t$, and thus the network will learn to output the same velocity vector regardless of the $t$-input. In other words,

$$\boldsymbol{v}_\theta(\boldsymbol{x}_\tau, t) = \boldsymbol{v}_\theta(\boldsymbol{x}_\tau, \tau), \quad \forall\, t \in [\tau, 1]. \tag{34}$$

This implies that the velocity predicted by the model at any target time $t$ coincides with the velocity at the starting time $\tau$, satisfying the *velocity consistency* condition:

$$\left. \frac{\partial f_\theta(\boldsymbol{x}_\tau, t)}{\partial t} \right|_{t=\tau} = \boldsymbol{v}_\theta(\boldsymbol{x}_\tau, \tau) = \boldsymbol{v}_\theta(\boldsymbol{x}_\tau, t). \tag{35}$$

$\square$

### A.3 PROOF OF COROLLARY 1

**Corollary 1** *Let $\Delta t$ be a small time increment and $f_\theta(\boldsymbol{x}_\tau, t)$ be a trajectory function that satisfies the following local consistency condition:*

$$f_\theta(\boldsymbol{x}_\tau, t) = f_\theta(\boldsymbol{x}_{\tau+\Delta t}, t), \quad \text{where } \boldsymbol{x}_{\tau+\Delta t} = f_\theta(\boldsymbol{x}_\tau, \tau + \Delta t). \tag{36}$$

*Then, for any $\tau < t_1 < t_2$, the function also satisfies the global* Self-Consistency *constraint:*

$$f_\theta(\boldsymbol{x}_\tau, t_2) = f_\theta(f_\theta(\boldsymbol{x}_\tau, t_1), t_2). \tag{37}$$

*Proof.* We aim to show that repeated application of the local condition leads to the global *Self-Consistency* property. Divide the interval from $\tau$ to $t_1$ into small steps of size $\Delta t$, such that:

$$\tau = \tau_0 < \tau_1 = \tau + \Delta t < \tau_2 = \tau + 2\Delta t < \cdots < \tau_n = t_1. \tag{38}$$

Using the local condition iteratively, we obtain:

$$f_\theta(\boldsymbol{x}_\tau, t_2) = f_\theta(\boldsymbol{x}_{\tau_1}, t_2) = f_\theta(\boldsymbol{x}_{\tau_2}, t_2) = \cdots = f_\theta(\boldsymbol{x}_{\tau_n}, t_2), \tag{39}$$

where each $\boldsymbol{x}_{\tau_i} = f_\theta(\boldsymbol{x}_\tau, \tau_i)$ for $i = 1, \ldots, n$.

In particular, $\boldsymbol{x}_{\tau_n} = f_\theta(\boldsymbol{x}_\tau, t_1)$, and hence:

$$f_\theta(\boldsymbol{x}_\tau, t_2) = f_\theta(f_\theta(\boldsymbol{x}_\tau, t_1), t_2), \tag{40}$$

which proves the global *Self-Consistency* constraint. $\square$

### A.4 PROOF OF PROPOSITION 2

**Proposition 2.** *Suppose the model $\theta$ is trained using the following loss:*

$$\mathcal{L}_{vec} = \left\| \boldsymbol{v}_\theta^*(\boldsymbol{x}_\tau, t_2) - \boldsymbol{v}_\theta^*(\boldsymbol{x}_{t_1}, t_2) \right\|_2^2. \tag{41}$$

*Then $\boldsymbol{v}_\theta^*$ corresponds to the instantaneous velocity at the terminal position, and such velocities are simultaneously guaranteed to remain self-consistent across decompositions.*

*Proof.* During training, $t_1$ is sampled arbitrarily over the trajectory and all such $t_1$ points are used to supervise $\boldsymbol{v}_\theta^*$. Therefore, at convergence, we have

$$\boldsymbol{v}_\theta^*(\boldsymbol{x}_{t_1}, t_2) = \boldsymbol{v}_\theta^*(\boldsymbol{x}_{t_2}, t_2), \tag{42}$$

for any $t_1$ along the trajectory.

By definition, $\boldsymbol{v}_\theta^*(\boldsymbol{x}_{t_2}, t_2)$ corresponds to the instantaneous velocity at the start position. That is,

$$\boldsymbol{v}_\theta^*(\boldsymbol{x}_{t_2}, t_2) = \boldsymbol{v}_\theta(\boldsymbol{x}_{t_2}, t_2) = \left. \frac{\partial f_\theta(\boldsymbol{x}_\tau, t)}{\partial t} \right|_{t=t_2}. \tag{43}$$

Combining the above equalities, we see that the self-consistency velocity loss ensures both:

1. **Terminal velocity agreement**: $v_\theta^*$ correctly predicts the instantaneous velocity of $f_\theta$ at $t_2$.

2. **Trajectory decomposition consistency**: velocities predicted from any intermediate point $x_{t_1}$ match the terminal velocity, i.e.,

$$v_\theta^*(x_{t_1}, t_2) = v_\theta^*(x_\tau, t_2).$$

Hence, minimizing $\mathcal{L}_{\text{vec}}$ guarantees that $v_\theta^*$ remains self-consistent across trajectory decompositions while accurately representing the instantaneous velocity at the terminal state.

$\square$

## B  EXPLANATION OF EQUATION 18

A natural way to strengthen the self-consistency constraint is to additionally enforce velocity consistency at $t_2$ from any starting point $t_1$ along the same trajectory. Formally, given $x_\tau$ and $x_{t_1}$ with $\tau < t_1 < t_2$, we require

$$v_\theta\left(\hat{x}_{t_2}^{(\tau)}, t_2\right) = v_\theta\left(\hat{x}_{t_2}^{(t_1)}, t_2\right), \tag{44}$$

where $v_\theta(x, t) = \left.\frac{\partial f_\theta(x,t)}{\partial t}\right|_t$ denotes the instantaneous velocity.

Concretely, the computation proceeds as follows:

$$x_{t_1} = f_\theta(x_\tau, t_1), \quad \hat{x}_{t_2}^{(\tau)} = f_\theta(x_\tau, t_2), \quad \hat{x}_{t_2}^{(t_1)} = f_\theta(x_{t_1}, t_2). \tag{45}$$

We then evaluate the instantaneous velocities at these two candidate terminal points:

$$v^{(\tau)} = v_\theta\left(\hat{x}_{t_2}^{(\tau)}, t_2\right), \quad v^{(t_1)} = v_\theta\left(\hat{x}_{t_2}^{(t_1)}, t_2\right). \tag{46}$$

The strengthened velocity-consistency loss is defined as

$$\mathcal{L}_{\text{vec}} = \left\| v^{(\tau)} - v^{(t_1)} \right\|_2^2. \tag{47}$$

**Computational Cost.** This strengthened constraint requires three forward evaluations of $f_\theta$ to obtain $x_{t_1}$, $\hat{x}_{t_2}^{(\tau)}$, and $\hat{x}_{t_2}^{(t_1)}$. These forward passes are already present in the original self-consistency computation and therefore do not introduce extra cost. The main additional overhead comes from the two velocity evaluations $v_\theta\left(\hat{x}_{t_2}^{(\tau)}, t_2\right)$ and $v_\theta\left(\hat{x}_{t_2}^{(t_1)}, t_2\right)$. In practice, this corresponds to gradient-based directional computations through the network, which are substantially more expensive than standard forward evaluations. As a result, enforcing velocity-consistency leads to significantly higher training overhead compared with enforcing value-consistency alone.

## C  SAMPLING

Our framework incorporates two model distributions, $p_\theta$ and $q_\theta$, enabling flexible and efficient sampling strategies. Following standard flow matching approaches, sampling from $p_\theta$ can be generated by solving the neural ODE defined by $v_\theta^{(p)}$ using numerical solvers (Chen et al., 2018). Alternatively, fast sampling is enabled by directly evaluating $q_\theta$ via the learned mapping $f_\theta$, which eliminates the need for iterative solvers. In addition, since $f_\theta$ is explicitly constructed with a Taylor-like expansion in time, its second-or third-order time derivative at the initial point is analytically accessible:

$$\left.\frac{\partial^2 f_\theta(x_\tau, t)}{\partial t^2}\right|_{t=\tau} = u_\theta(x_\tau, \tau), \quad \left.\frac{\partial^3 f_\theta(x_\tau, t)}{\partial t^3}\right|_{t=\tau} = w_\theta(x_\tau, \tau) \tag{48}$$

This property allows us to employ second-order ODE solvers for sampling from $p_\theta$, potentially achieving higher accuracy and efficiency than conventional first-order methods used in prior work. We summarize the sampling procedures from both $q_\theta$ (via few-step generation) and $p_\theta$ (via second-order numerical integration) in Algorithm 4 and Algorithm 5 respectively.

**Algorithm 4** Sampling from $q_\theta$

**Input:** model $\mathcal{F}_\theta$, sequence of time points: $0 = t_1 < t_2 < \cdots < t_{N-1} < t_N = 1$
Sample inital noise $\boldsymbol{x}_{t_1} \sim \mathcal{N}(\mathbf{0}, \boldsymbol{I})$
**for** $i = 1$ **to** $N - 1$ **do**
  $\boldsymbol{x}_{t_{i+1}} \leftarrow f_\theta(\boldsymbol{x}_{t_i}, t_{i+1})$
**end for**
$x \leftarrow \boldsymbol{x}_{t_N}$
**Output:** $x$

**Algorithm 5** Second-order ODE Solver

**Input:** model $\mathcal{F}_\theta$, sequence of time points: $0 = t_1 < t_2 < \cdots < t_{N-1} < t_N = 1$
Sample inital noise $\boldsymbol{x}_{t_1} \sim \mathcal{N}(\mathbf{0}, \boldsymbol{I})$
**for** $i = 1$ **to** $N - 1$ **do**
  $\boldsymbol{v}, \boldsymbol{u} \leftarrow f_\theta(\boldsymbol{x}_{t_i}, t_i)$
  $\boldsymbol{x}_{t_{i+1}} \leftarrow \boldsymbol{x}_{t_i} + (t_{i+1} - t_i)\boldsymbol{v} + \frac{1}{2}(t_{i+1} - t_i)^2 \boldsymbol{u}$
**end for**
$x \leftarrow \boldsymbol{x}_{t_N}$
**Output:** $x$

# D    EXPERIMENTAL DETAILS

## D.1    TRAINING USING THE DEFAULT CONFIGURATION OF RECTIFIED FLOW

We use the Adam optimizer with $\beta_1 = 0.9$, $\beta_2 = 0.999$, and a learning rate of $1 \times 10^{-4}$. The TCF models are trained for 1M steps with a batch size of 512 across 2 GPUs. We adopt cosine learning rate decay and apply exponential moving average (EMA) with a decay rate of 0.9999. Most architectural configurations follow the RF baseline, with the following key modifications:

- Unlike standard flow matching models, our framework requires an additional time input and produces an extra output. To accommodate this, we introduce a dedicated time embedding layer that conditions the model on both the reference time $\tau$ and the target time $t$.

- The output dimension of the TCF model is set to 6 for Alg-A, B and 9 for Alg-C instead of the standard 3 (for RGB images), which we interpret as a concatenation of two or three 3D vectors.

Additionally, we apply a dropout rate of 0.2 throughout all our experiments.

For the consistency training (CT) model, we follow the exact architecture used in RF. The CT model is trained for 2M steps on 2 GPUs with a total batch size of 512.

## D.2    EXPLORING DESIGN CHOICES

To identify the optimal design choices for our model, we conduct a series of ablation studies, presented in Table 4. We adopt the same network architecture, hyperparameters, and data augmentation strategies as EDM (Karras et al., 2022) for CIFAR-10 and ImageNet-64, ensuring comparability and robustness of our experiments. The resulting model sizes are 55M and 296M parameters, respectively. All models in these experiments are built upon a base $p_\theta$ model that achieves an FID of 2.10 on CIFAR-10. The auxiliary variable $\tau$ is sampled from a log-normal distribution, and all evaluations are performed under a fixed number of function evaluations (NFE = 2). Each model is trained for a total of 400k steps, ensuring convergence while allowing for a controlled comparison of different factors, including the distributions of $t_1$ and $t_2$, time conditioning, training weights, loss types, and dropout rates.

## D.3    EXPLORING DIFFERENT TRAINING STRATEGIES FOR TCF

We explore different training strategies for TCF to evaluate their impact on performance:

1. **Training from scratch:** TCF is trained entirely from random initialization.

2. **Initialization with a pretrained flow matching model:** The key distinction of TCF from standard flow matching or diffusion models lies in the inclusion of an additional time step input and one or two auxiliary outputs. To ensure stable training, we initialize the linear mapping layers corresponding to the new input and output to zero. This allows the network to start training with a well-performing baseline, and thanks to the presence of the flow matching loss, the performance of the pretrained components remains largely unchanged throughout training.

3. **Distillation from a pretrained flow matching model:** Similar to the previous setup, we initialize the additional input and output layers to zero and perform knowledge distillation from a pretrained

| **1. Distribution of $t_1$ and $t_2$** | |
|---|---|
| Setting | FID |
| $t_2 \sim U(\tau, 1), t_1 \sim U(\tau, t_2)$ | 3.65 |
| $t_2 \sim U(\tau, 1), t_1 = 0.5(\tau + t_2)$ | 3.75 |
| $t_2 \sim 1 - (1 - U(\tau, 1))^{1.5}, t_1 \sim U(\tau, t_2)$ | **3.20** |
| $t_2 \sim 1 - (1 - U(\tau, 1))^2, t_1 \sim U(\tau, t_2)$ | 3.40 |
| **2. Training weights $(\lambda_1, \lambda_2, \lambda_3)$** | |
| Setting | FID |
| 1,1,1 | 3.65 |
| 1,2,1 | 3.55 |
| 1,5,1 | **3.10** |
| 1,5,2 | 3.20 |
| 1,5,0.5 | 3.15 |
| **3. Time conditioning** | |
| Condition | FID |
| $\tau, t$ | 3.65 |
| $t$ | 8.31 |
| $\tau, t - \tau$ | **3.55** |
| **4. Loss types $(L_{\mathrm{FM}}, L_{\mathrm{consist}}, L_{\mathrm{vec}})$** | |
| Loss | FID |
| L2, L2, L2 | **3.65** |
| L2, Pseudo-Huber, Pseudo-Huber | 3.68 |
| Pseudo-Huber, Pseudo-Huber, Pseudo-Huber | 3.92 |
| **5. Dropout rate** | |
| Dropout | FID |
| 0.13 | 3.65 |
| 0.20 | **3.55** |
| 0.30 | 3.63 |
| 0.00 | 4.12 |

Table 4: Ablation studies exploring design choices for our model. Each section reports FID under specific variations of distributions, training weights, time conditioning, loss types, and dropout rates.

flow matching model. During this process, we substitute the reference vector field (e.g., $\boldsymbol{x}_1 - \boldsymbol{x}_0$) with the vector field generated by the pretrained model, allowing the student network to directly learn from the teacher's trajectory.

4. **Decoupled training of flow matching module ($v_\theta$) and other components ($\boldsymbol{u}_\phi, \boldsymbol{w}_\phi$):** To reduce training time, we directly use a trained flow matching model to predict the velocity $\boldsymbol{v}_\theta$, freezing its parameters during the training of the auxiliary components ($\boldsymbol{u}_\phi$ and $\boldsymbol{w}_\phi$). At inference time, we combine $\boldsymbol{v}_\theta$ with the learned auxiliary components $\boldsymbol{u}_\phi$ and $\boldsymbol{w}_\phi$ to generate the final trajectories, enabling consistent and efficient sampling.

For strategies 2, 3 and 4, we employ a pretrained EDM model as the initialization. For strategies 3, 4, since the flow matching part is deterministic, we adopt a modified time sampling strategy: we first sample $D \sim U(0, 1)$, then $\tau \sim U(0, 1 - D)$, and set $t_2 = \tau + D$. This design increases the expected time interval during training, which facilitates better few-step inference performance.

| Training Strategy | Training Steps | FID |
|---|---|---|
| 1. Training from scratch | 800k | 2.95 |
| 2. Initialization with pretrained flow matching | 600k | 2.93 |
| 3. Distillation from pretrained flow matching | 200k | 2.23 |
| 4. Decoupled training of $v_\theta$ and $(u_\phi, w_\phi)$ | 200k | 2.18 |

Table 5: Comparison of training steps and FID for different TCF training strategies. FID is evaluated at NFE=2 for all strategies, except strategy 4, which effectively uses double NFE and roughly twice the number of model parameters due to decoupled training.

## E   RELATED WORK: DISTILLATION FOR FAST SAMPLING

To accelerate sampling in diffusion and flow matching models, various model distillation strategies have been proposed. A common approach is to use a pre-trained teacher model to generate synthetic datasets, which are then used to supervise a fast student model (Luhman & Luhman, 2021; Liu et al., 2023; Lee et al., 2024; Kim et al., 2025). However, these methods often suffer from performance degradation due to the limited quality or diversity of the synthetic data. Another line of work adopts progressive distillation, where sampling trajectories are shortened in multiple stages (Salimans & Ho, 2022). While this reduces computation, it tends to introduce cumulative approximation errors that affect final sample quality. More recent approaches employ consistency-based objectives, such as trajectory alignment or self-consistency constraints, to distill models in a single stage, avoiding reliance on synthetic data while achieving faster sampling with improved fidelity (Song et al., 2023). A different class of methods learns auxiliary networks to approximate the score function or its implicit distribution, enabling direct sampling but requiring careful design and tuning of additional components (Luo et al., 2024). Unlike all the approaches mentioned above, our method avoids the need for teacher models, synthetic datasets, or auxiliary score predictors. Instead, we jointly train a base model $p_\theta$ and a fast sampler $q_\theta$, where alignment between the two distributions is enforced through integration-free consistency constraints. This design allows for efficient training and accurate, solver-free sampling in a fully self-contained framework.

## F   LIMITATIONS

While our method achieves high-quality image generation, several limitations remain. First, although our approach dramatically reduces the number of sampling steps, the resulting sample quality still slightly lags behind the best-performing consistency models in terms of FID. This performance gap could potentially be addressed in future work by exploring more effective time schedules and designing improved loss functions that better capture the alignment between the model distributions. Second, our experiments are primarily conducted on standard datasets for unconditional or class-conditional image generation. Extending the method to more complex, multi-modal domains, such as text-to-image synthesis or video generation, may require additional architectural and algorithmic innovations. Lastly, jointly optimizing two model distributions introduces extra complexity in training, including the need for careful tuning of the loss coefficient to maintain a proper balance between the objectives. This may lead to increased training overhead. Future research may explore more adaptive training strategies, alternative fast-sampling parameterizations, and broader extensions of the framework to diverse data modalities and tasks.

## G   VISUALIZATION RESULTS

We present the visualization results on the next pages.

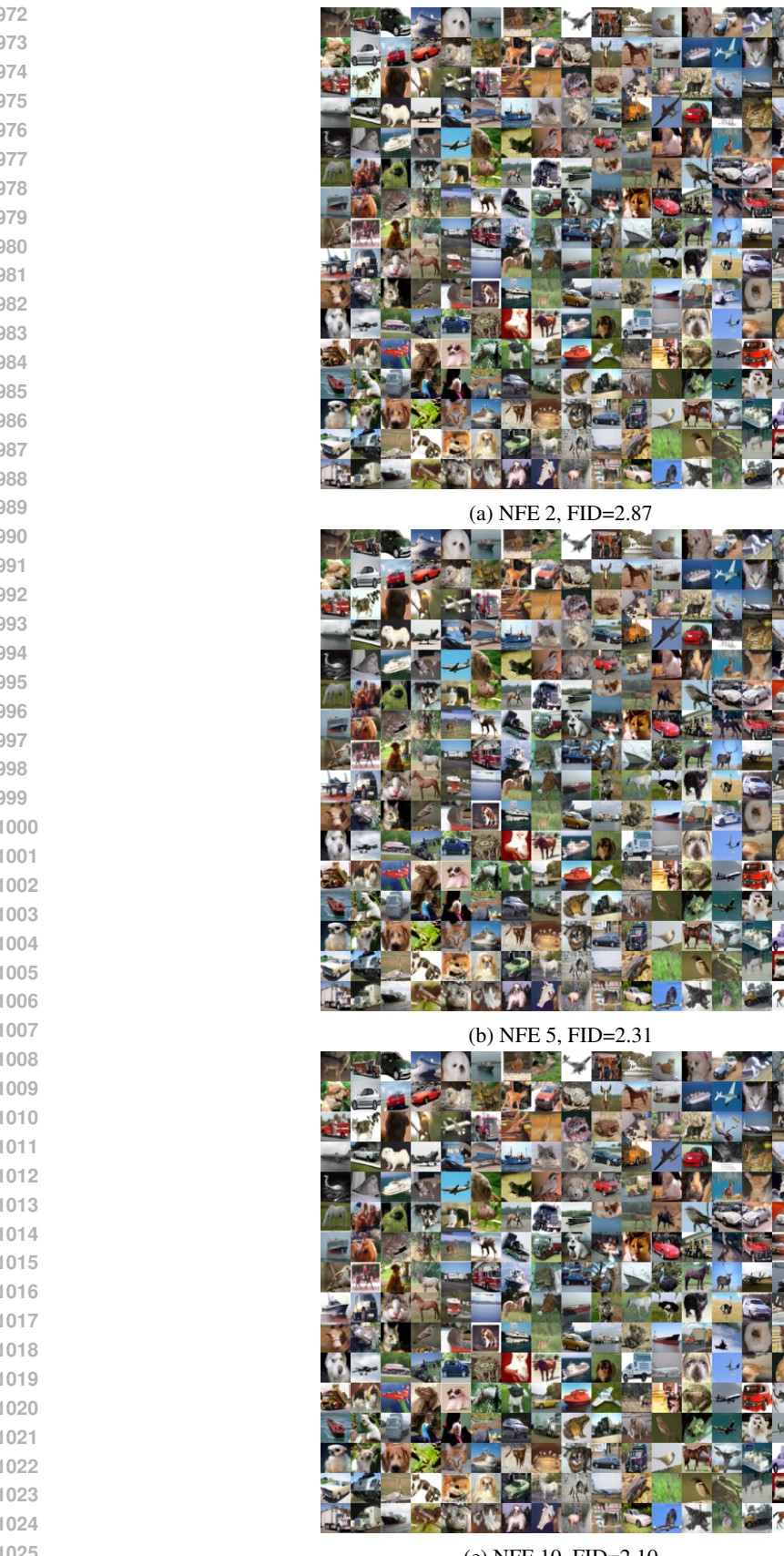

(a) NFE 2, FID=2.87

(b) NFE 5, FID=2.31

(c) NFE 10, FID=2.10

Figure 3: Unconditional CIFAR-10 results, Alg A

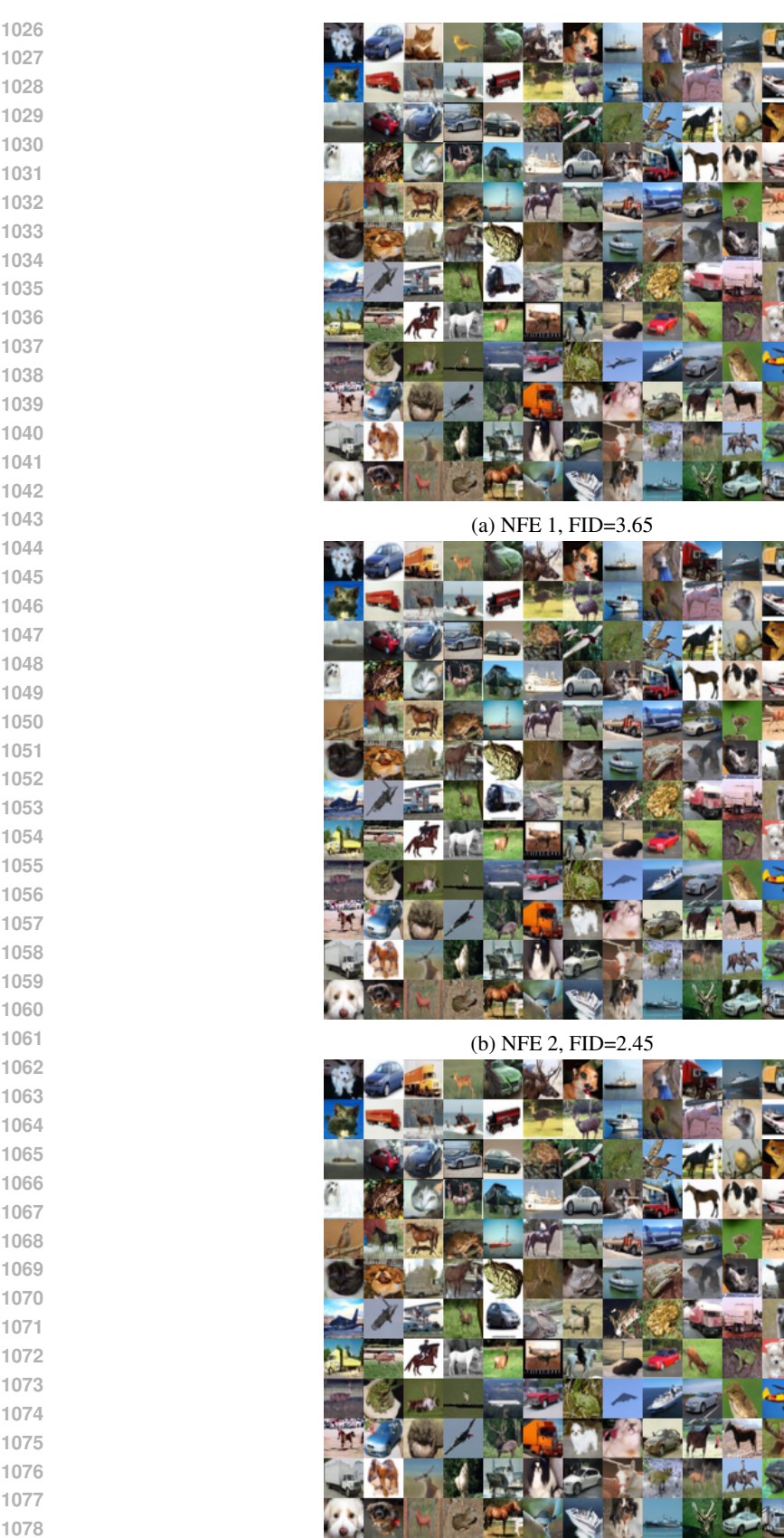

(a) NFE 1, FID=3.65

(b) NFE 2, FID=2.45

(c) NFE 5, FID=2.28

Figure 4: Unconditional CIFAR-10 results, Alg C

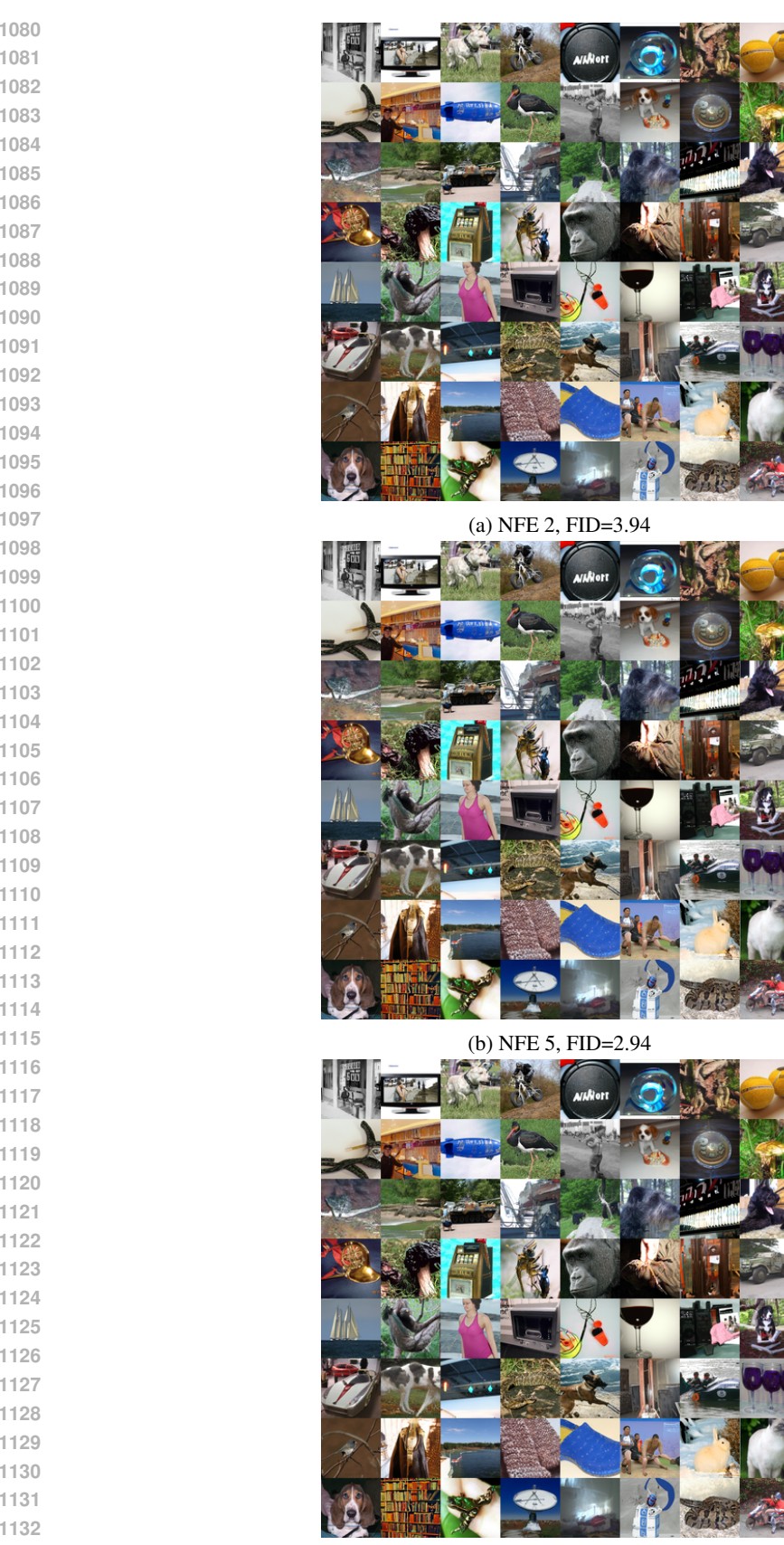

(a) NFE 2, FID=3.94

(b) NFE 5, FID=2.94

(c) NFE 10, FID=2.18

Figure 5: Results on ImageNet $64 \times 64$ (class-conditional), Alg C.

