# OpenReview forum: "Trajectory-Consistent Flows: Enabling Fast Sampling for Flow Matching Models"
_ICLR.cc/2026/Conference — ICLR 2026 Conference Withdrawn Submission_

### Official Review · Reviewer_VCmg · 2025-10-29

**Soundness:** 3
**Presentation:** 1
**Contribution:** 3
**Rating:** 4
**Confidence:** 5

**Summary:**

This paper proposes a method to jointly learn the velocity and solution of PF-ODE. Experiments are conducted on CIFAR-10 and ImageNet-64.

**Strengths:**

Improving discrete-time consistency models (which do not rely on the Jacobian-vector product) and their variants is an important direction. The proposed idea, including the parallel prediction of the trajectory function and the velocity, can be useful for training the efficiency of these models.

**Weaknesses:**

My biggest concern is the positioning of this work, as many existing concepts are proposed as contributions in Sec. 3. Modeling both the drift and the solution of PF-ODE with a single neural network is not new; it has been done by CTM, flow-map matching, and related ideas. Eq. 6 and 8 are well known and already present in these works. It would be better to have a separate background section on CTM and related ideas to isolate the novel contribution of this work, and maybe present Eq. 6 and 8 as background.

Due to this unclear positioning, what appears to be not new is emphasized throughout the paper, and as a result, what appears to be new does not get emphasized properly.

* To my knowledge, what appears to be new is Eq. (14), where they use f itself to generate the teacher input at t2 (Algorithm A). This is usually done by using $v_\theta$ (or a Monte Carlo estimation of the ground-truth velocity, as done in consistency training) in prior work; this is referred to as Algorithm B in the paper. However, Algorithm B turns out to be more efficient and performs on par with Algorithm A (as shown in Fig. 2), so whether Eq. (14) is an improvement is not clear.
* Another novelty is that Eq. 9 can produce f and v using only one forward pass due to its multi-head architecture. Prior work normally requires two forward passes.

I'd suggest 1) properly discussing the prior work on learning the trajectory function of PF-ODE, and 2) upon that, presenting the contributions (e.g., a bootstrapping method (Eq. 14) and parallel prediction (Eq. 9)). I can raise my score if the above concerns are resolved.

**Questions:**

Is it correct to say u captures the second-order derivative? It's learning a residual between the true trajectory function and first-order approximation (i.e., the sum of all higher-order terms). For this reason, I'm not sure if this can be plugged into a second-order ODE solver.

---

### Official Review · Reviewer_Js8Q · 2025-10-30

**Soundness:** 2
**Presentation:** 2
**Contribution:** 2
**Rating:** 2
**Confidence:** 4

**Summary:**

This paper proposes Trajectory Consistent Flows (TCF), which jointly learns a flow model, which learns the flow velocity, and a trajectory model, which translates samples from one timepoint to another timepoint along the flow probability-flow ODE (PF-ODE) with a single function evaluation. Flow and trajectory models are combined into a single neural net, TCF, whose function value returns the jump along the flow PF-ODE, and time-derivative value returns the flow velocity. TCF is optimized with a combination of self-consistency loss (similar to the loss for shortcut models) which distills PF-ODE trajectories, and velocity consistency loss which matches TCF time-derivatives to flow velocities. TCF demonstrates competitive performance on the task of CIFAR10 and ImageNet-64 generation.

**Strengths:**

- **[S1] This paper is significant in the aspect that it provides several design choices which improve the generative performance of shortcut models.** Specifically, the authors demonstrate that by using a Taylor-expansion based parametrization Eq. (9) along with careful design choices such as time distributions, time conditioning, weighting coefficients for losses (Section 4.3), one can improve FID scores for shortcut models (Table 1).

**Weaknesses:**

- **[W1] TCF lacks conceptual novelty, in the sense that it is very similar to shortcut models.** Specifically, TCF is also trained with a combination of flow matching and shortcut learning losses. The only difference between TCF and shortcut model is the choice of parametrization: TCF directly models the jump along PF-ODE trajectories whereas shortcut model learns the direction from one PF-ODE state to another.

- **[W2] TCF lacks practical significance, as there are fast flow methods with similar or cheaper training costs but better generative performance.** For instance, the authors claim in lines 427-429 that baselines such as truncated consistency models (TCMs) or sCMs come at the cost of increased training or inference complexity. However, I would like to argue the contrary: TCM requires similar per-iteration cost, whereas sCM has cheaper per-iteration cost compared to TCF during training.

  Specifically, the results in Tables 2 and 3 use Alg-C for TCF, which requires three forward passes and one backward pass of the network per iteration. TCM also requires three forward passes and one backward pass of the network per training iteration. While sCM requires JVP computation they use forward-mode automatic differentiation, which is significantly cheaper than backpropgation. Hence, as written in Discussions and Limitations of [1], the cost of sCM training per iteration is similar to two forward passes and one backward pass, so it is cheaper than TCF.

[1] Simplifying, Stabilizing and Scaling Continuous-Time Consistency Models, ICLR, 2025

**Questions:**

- **[Q1] The precise training setup for TCFs in Tables 2 and 3 are unclear.** Are the models initialized from a pre-trained diffusion / flow model, or are they trained from scratch? Also, in lines 402-404, the authors write that they increase the batch size from 512 to 2048, but in  Appendix D.1, the authors write they use batch size 512. With experimental details scattered across the paper, it is difficult to fairly compare TCF against the baselines. Please describe the complete training setup (e.g., network initialization, number of training iterations, batch size, etc.) for the models in Tables 2 and 3.

---

### Official Review · Reviewer_UVaF · 2025-11-05

**Soundness:** 2
**Presentation:** 1
**Contribution:** 2
**Rating:** 2
**Confidence:** 4

**Summary:**

The paper introduces Trajectory Consistent Flows (TCF), a new framework designed to accelerate the sampling process in flow-based generative models. TCF jointly optimizes a standard flow model and a fast-sampling surrogate model that learns the trajectory of the Probability Flow ODE. The key idea is to maintain trajectory consistency between these two models so that the surrogate can effectively approximate multi-step transitions without losing fidelity. Conceptually, this approach shares similarities with FlowMap, Shortcut based models.
To make the surrogate model computationally efficient, the author parameterizes it through a second-order (or third-order) Taylor expansion. This expansion neatly decomposes the model into two components: the standard flow term and a remainder term capturing the deviation of the trajectory from the base flow. Both components share the same neural backbone but are separated through distinct output heads.
The model is trained with three objectives that, together, enforce trajectory consistency: (1) boundary conditions, (2) self-consistency (similar to the Shortcut Model), and (3) velocity consistency (analogous to Flow Matching objectives).
Experimental results show that TCF achieves competitive performance compared to other fast-sampling models on CIFAR-10 and ImageNet 64.

**Strengths:**

**[S1] Theoretical formulation and rigor**

The paper provides well-structured theoretical evidence. The author proposes necessary conditions for trajectory consistency and offers rigorous proofs demonstrating that TCF satisfies them. The definitions and organization of trajectory consistency and its conditions are clear and insightful, providing valuable conceptual guidance for future research in the field.

**Weaknesses:**

**[W1] Limited novelty**

Although TCF is theoretically neat, the level of novelty is somewhat limited. Several ideas such as self-consistency objective were already introduced in earlier methods such as the Shortcut Model. Especially, The training process in Algorithm B closely resembles that of MeanFlows [1], where the Jacobian–vector product is approximated through finite differences. Although the Taylor expansion–based parameterization is a novel element, it may be viewed as an incremental contribution rather than a breakthrough.

**[W2] Insufficient comparison with related flow-map models**

Since TCF explicitly maps a sample from one timestep to another, it shares strong conceptual ties with flow-map models such as Mean Flows [1] and AYF [2]. However, the comparisons to these models are limited and somewhat superficial.
Moreover, Table 1 and Figure 2 compares TCF only with CT and the Shortcut Model, leaving out several recent methods that directly target the same acceleration problem. Including more recent baselines such as ECM [3], sCM [4], and IMM [5] would offer a more convincing evaluation.

**[W3] Modest empirical performance**

While TCF achieves competitive results, it fails to demonstrate clear superiority over leading models such as ECM, sCM, TCM [6], MeanFlow, and IMM [5]. The author argues that TCF maintains advantages such as a simpler training process and avoidance of expensive Jacobian–vector products. However, this justification is not entirely convincing since ECM and IMM already enjoy similar benefits while achieving better performance.
Thus, even if TCF offers some efficiency, the absence of a significant performance gain weakens its practical contribution.

**[W4] Inconsistency in terminology (“two-stage” vs. “single-stage”)** (minor)

There is a potential inconsistency between lines 392 (“two-stage strategy”) and 430 (“single-stage training process”). Although these may refer to different contexts, the terminology could confuse readers. The paper would benefit from a clearer explanation of what “stage” means in each case.

[1] Mean Flows for One-step Generative Modeling, NeurIPS 2025

[2] Align Your Flow: Scaling Continuous-Time Flow Map Distillation, NeurIPS 2025

[3] Consistency Models Made Easy, ICLR 2025

[4] Simplifying, Stabilizing and Scaling Continuous-Time Consistency Models, ICLR 2025

[5] Inductive Moment Matching, ICML 2025

[6] Truncated Consistency Models, ICLR 2025

**Questions:**

**[Q1] Mismatch between mathematical notations and Table 4**

Equations (4)–(22) define the network $f_\theta$ as conditioned only on the destination timestep. However, Table 4 describes design choices where the network is conditioned on two timesteps. Was this an intentional extension of the formulation? If so, does the theoretical framework — including the proofs of trajectory consistency — still hold under dual-timestep conditioning?

**[Q2] Ablation on the number of sampling steps across different methods**
It would be helpful to analyze how the generative performance of other models such as MeanFlows, IMM, and ECM varies with the number of sampling steps. Since TCF aims to accelerate sampling while preserving sample quality, understanding how performance degrades (or remains stable) under fewer sampling steps is crucial for fair comparison.

---

### Author Response · Authors · 2025-12-04
**Withdrawal Comment**

We sincerely thank all reviewers for the time, effort, and constructive feedback provided during the review process.

After careful consideration, we have decided to **withdraw our submission**.

Recently, we have made **substantial new progress** on this research direction, which requires major revisions and restructuring of the current paper. To appropriately incorporate these new developments and ensure a more complete and rigorous presentation, we believe it is best to withdraw the current version and resubmit a significantly improved manuscript in the future.

We greatly appreciate the reviewers’ insightful comments, which will guide the next iteration of this work.

Thank you again for your valuable feedback and understanding.

**Best regards,**
*The Authors*

---

### Note · Authors · 2025-12-04

**Comment:**

We sincerely thank all reviewers for the time, effort, and constructive feedback provided during the review process.

After careful consideration, we have decided to **withdraw our submission**.

Recently, we have made **substantial new progress** on this research direction, which requires major revisions and restructuring of the current paper. To appropriately incorporate these new developments and ensure a more complete and rigorous presentation, we believe it is best to withdraw the current version and resubmit a significantly improved manuscript in the future.

We greatly appreciate the reviewers’ insightful comments, which will guide the next iteration of this work.

Thank you again for your valuable feedback and understanding.

**Best regards,**
*The Authors*

**Withdrawal Confirmation:**

I have read and agree with the venue's withdrawal policy on behalf of myself and my co-authors.